# Ten Principles for Bird-Friendly Forestry: Conservation Approaches in Natural Forests Used for Timber Production

**Nico Arcilla [1,2,*] and Māris Strazds [3]**

1   International Bird Conservation Partnership, Storövägen 13, 14142 Huddinge, Sweden
2   Center for Great Plains Studies, University of Nebraska-Lincoln, 1155 Q St., Lincoln, NE 68588, USA
3   Laboratory of Ornithology, Institute of Biology, University of Latvia, 1050 Riga, Latvia
*   Correspondence: nico.arcilla@aya.yale.edu

**Simple Summary:** Most of the world's natural forests are subject to logging operations, many of which are highly detrimental to forest birds and other wildlife. Considerable scientific research has investigated approaches in sustainable forest management, which strives to mitigate forest degradation and wildlife loss in forests subject to logging operations and is fostered through conservation legislation and incentives in many areas. We reviewed relevant research to synthesize 10 principles that support bird conservation in forests subject to logging operations: (1) protect and enhance vertical structure through uneven-aged silviculture; (2) leave abundant dead wood in different decay stages; (3) maintain residual large green trees; (4) create and maintain sufficient amounts of uncut reserves and corridors; (5) maximize forest interior and minimize fragmentation by retaining large contiguous forest tracts; (6) maintain buffers along streams, rivers, wetlands, and known nesting areas; (7) maintain horizontal stand structure and vegetation diversity through canopy gaps; (8) extend the temporal scale of logging cycles; (9) minimize post-logging disturbance to forests, particularly during the bird breeding season; and (10) manage for focal species and guilds. Foresters can customize these principles in management plans to improve the bird conservation value of production forests, setting quantitative objectives to be measured using bird monitoring data.

**Abstract:** Bird–forestry relationships have been the subject of research and conservation initiatives for decades, but there are few reviews of resulting recommendations for use by forest managers. We define "bird-friendly forestry" as forest management that applies recommendations from research seeking to reconcile logging with bird conservation in natural forests used for timber production. We reviewed relevant studies to synthesize 10 principles of bird-friendly forestry: (1) protect and enhance vertical structure through uneven-aged silviculture; (2) leave abundant dead wood in different decay stages; (3) maintain residual large green trees; (4) create and maintain sufficient amounts of uncut reserves and corridors; (5) maximize forest interior by retaining large contiguous forest tracts in landscapes with sufficient functional connectivity; (6) maintain buffers along streams, rivers, and wetlands cultural and urban landscapes; (7) maintain horizontal stand structure and enhance vegetation diversity by creating canopy gaps; (8) extend the temporal scale of logging cycles; (9) minimize post-logging disturbance to forests, particularly during the bird breeding season; and (10) manage for focal species and guilds. These principles may serve as guidelines in developing bird-friendly management plans customized for regional priority species, with a clearly articulated vision and quantitative objectives through which success can be measured.

**Keywords:** silviculture; natural disturbance emulation; indicator species; Europe; North America; boreal forests; temperate forests

## 1. Introduction

Most of the world's forests are used for the production of timber and other economic commodities, the demand for which is expected to grow [1]. Forest degradation and loss

associated with logging operations are detrimental to many birds and other species [2–4], and forest management practices that maintain forest biodiversity are urgently needed [5]. Understanding the responses of forest birds to timber harvest operations is fundamental to conservation and has been the subject of many studies, but there are surprisingly few reviews of their findings [6–8]. In many timber production forests, damage from logging operations including clearcutting, thinning, and soil scarification drive biodiversity declines [9–11]. Europe has the highest proportion of the world's forests dedicated to the production of wood and other commodities [1]. More than 95% of the natural forests of the Fennoscandian countries of Norway, Sweden, and Finland have already been logged [5], and monitoring data show that forest bird populations all three countries have been declining for the last three decades [2]. The expanding timber frontier has led to the elimination of unlogged forest in Finland [1] and declines in the last intact old-growth forests in Sweden [12]. Although the exploitation of North American forests has a shorter history than that in Europe, logging operations have also changed North American forests and their bird communities, such as the conversion of southern Québec's boreal forest from mixed to deciduous forest [4,13,14]. In addition, at least half (50–60%) of remaining tropical forests, which harbor the world's highest levels of terrestrial biodiversity, are dedicated to timber production, with corresponding declines in forest birds [10,11,15].

Mitigating bird declines associated with logging in natural forests is a goal towards which much research and legislation has been directed, particularly in Europe and North America [2,6–9,15]. Here, we define natural forest as land that has likely been historically continuously wooded with native flora, featuring natural composition and regeneration [16]. Harvest operations to remove timber from natural forests can be placed into broad categories according to the extent to which the canopy is removed, their distribution over the forest area, and whether regenerating forest consists of even-aged or uneven-aged stands of trees [17]. Even-aged silviculture includes harvesting trees through clearcutting or other methods that remove all trees and forest canopy; forest regenerating after timber harvest operations is considered even-aged if the difference in ages of the oldest and youngest trees does not exceed 20% of the length of rotation (e.g., 8 years' difference for a 40-year rotation) [17,18], and periodic commercial thinning may be conducted to maintain the density and composition of desired timber trees. Uneven-aged silviculture, in which three or more age classes of trees are represented within forest stands, promotes more diverse forests through selective logging of trees or stands of particular species and age classes on routine cutting cycles such as 20, 40, or 70 or more years [17,18].

Effective biodiversity conservation cannot coexist with maximum sustained yield timber production [19–21]. In areas where the production of wood, paper, and other human commodities from forests plays an important economic role but threatens conservation objectives, policy makers, forest owners, and land managers require general principles to guide management decisions. In the search for alternatives to conventional logging operations, many scientists and foresters have experimented with approaches designed to mimic natural disturbance regimes [22–24]. Natural forests are dynamic ecosystems, changing over time as disturbances such as wildfires, storms, and insect infestations kill individual or small groups of trees, producing light gaps and dead wood [25–27]. Such events have cascading effects on other forest species, in which ecological succession following disturbances supports a range of birds and other wildlife [7,10,28,29].

In this paper, we present a concise list of recommendations for bird-friendly forestry, which we define here as forest management that seeks to reconcile logging with bird conservation in natural forests used for timber production (hereafter, production forests). Improving bird conservation in production forests has been the subject of many research and extension publications but few comprehensive reviews [5–8,30–37]. Because particular management decisions and actions depend on specific landscape contexts, species, and goals, we sought to identify general principles that can be customized into management guidelines [38]. We drew on an abundance of studies of bird–forestry relationships, particularly in boreal and temperate forests in Europe and North America, where the majority

of studies have taken place [2,8,9,39–48], as well in tropical forests [10,11,15,49,50]. The world's most extensive terrestrial biome, circumboreal forest includes arctic, subarctic, and northern regions dominated by a cold climate, short growing season, and a small number of tree genera [51]. Between boreal and tropical biomes, temperate forests are characterized by mid-latitudes and four seasons. Around the equator, tropical forests have warm temperatures year-round and the highest terrestrial biodiversity on earth [10,15].

Although decades of research have informed hundreds of scientific articles on the effects of production forest management on wildlife, including birds, but there has been little consensus on their applications or broad considerations to inform conservation [38]. We used a modified Delphi technique [52] to assess the current state of knowledge on forest management and bird conservation at a 2018 University of Latvia symposium and 2019 European Ornithologists' Union meeting. Here, we briefly review research on birds as indicators of forest conservation status as well as strategies to mitigate forest bird declines through alternatives to conventional industrial logging, including uneven-aged silviculture, continuous cover forestry, retention forestry, and natural disturbance-based management [30–36]. We then present 10 principles of bird-friendly forestry, which can serve as a starting point for forest landowners and managers seeking guidance in developing management plans that balance timber harvest with bird conservation. Species and forest conservation strategies must be area-specific and will ideally be the focus of a clearly articulated conservation plan with measurable objectives [38]. Data from forest bird monitoring programs can inform such plans, and monitoring programs can be customized for particular areas, species, and objectives [2,6–8,20,48].

## 2. Forest Birds as Indicators of Biodiversity and Environmental Change

Birds play crucial roles in forest structure and function as seed dispersers, pollinators, predators, prey, and ecosystem engineers, and thus serve as useful indicators of forest biodiversity and environmental change [53–62]. Although many birds persist in natural forests used for timber production, conservation areas and actions are critical to maintaining populations of many forest bird species. Integrating forest inventory and bird survey data are essential to inform conservation efforts, and a best practice guide is available for wild bird monitoring schemes that can be customized for any location [63]. The Pan-European Common Bird Monitoring Scheme (PECBMS) uses common birds as indicators of the general state of nature using large-scale and long-term monitoring data on changes in breeding populations across Europe [64]. Country-specific indicator data are available on forest birds in many cases, such as the Finnish Common Forest Bird Indicator [65]. A program of the U. S. Fish and Wildlife Service and its collaborators, Partners in Flight assesses the population trends of 448 species of land birds in the United States and Canada, and its most recent assessment documents substantial declines in many species [66]. In addition, the IUCN Red List provides valuable, relevant data on species population trends and threats [67], which the European Environment Agency uses as a basis for designating the conservation status of forest birds and other wildlife [68]. The Committee on the Status of Endangered Wildlife in Canada plays a similar role in Canada [69], and the U.S. Fish and Wildlife Service manages listings under the Endangered Species Act in the United States [70]. The online collection of species accounts, "Birds of the World," provides extensive details and references on birds' life histories, including conservation status and the effects of human activities [71].

Forest bird population declines over the last decades have included many European resident and American migratory species ([5,7,66]; Figure 1). Since 1980, common forest birds in Europe have declined by an average of 10%, with declines of 20% in northern Europe, 9% in central, eastern, and southern Europe, and increases of 1% in western Europe [64]. In North America, forest generalist birds have declined by 18%, boreal forest birds have declined by 33%, eastern forest birds have declined by 17%, and western forest birds have declined by 30% since 1970 [66,72]. European resident species that are negatively affected by logging include the Western Capercaillie (*Tetrao urogallus*), Hazel

Grouse (*Tetrastes bonasia*), Eurasian Three-toed Woodpecker (*Picoides tridactylus*), Lesser Spotted Woodpecker (*Dryobates minor*; Figure 1), Pine Grosbeak (*Pinicola enucleator*), Siberian Jay (*Perisoreus infaustus*), Siberian Tit (*Parus cinctus*), and Willow Tit (*Poecile montanus*), which have declined significantly since 1980 [7,64,71–74]. American migratory species that are negatively affected by logging include the Eastern Whip-poor-will (*Antrostomus vociferus*), Bicknell's Thrush (*Catharus bicknelli*), Cerulean Warbler (*Setophaga cerulea*-72%), Canada Warbler (*Cardellina canadensis*-62%) Wilson's Warbler (*Cardellina pusilla*-57%), Blackpoll Warbler (*Setophaga striata*-92%), and Purple Finch (*Haemorhous purpureus*-47%), which have declined between 47–92% since 1970 [66,69,71,75].

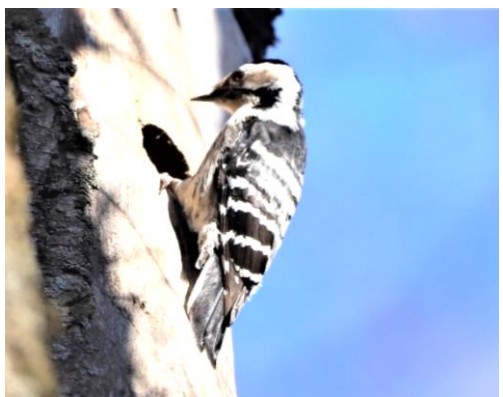 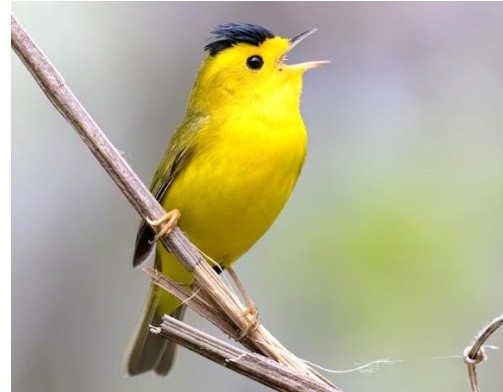

**Figure 1.** A European resident that is negatively affected by logging operations, the Lesser Spotted Woodpecker (*Dryobates minor*; left) has declined by 80% in boreal forests since 1980 [64]. An American migratory species that is negatively affected by logging operations, Wilson's Warbler (*Cardellina pusilla*; right) has declined by 57% in boreal forests since 1970 [66]. Lesser Spotted Woodpecker photo by Nico Arcilla; Wilson's Warbler photo by Dan Marks.

Particularly vulnerable to negative impacts of logging operations are avian guilds and species that rely on large, mature trees for breeding, notably woodpeckers and other cavity-nesting birds [2,4–7,21,76] as well as raptors and other canopy-nesting birds [2,4,7,8,77–80]. Many cavity nesters use snags, including dying, dead, and rotten trees, for nesting and foraging, and logging and snag removal can thus reduce or eliminate nesting and foraging opportunities [21,76,81,82]. Logging has been implicated in the declines of the Middle Spotted Woodpecker (*Dendrocoptes medius*) in Sweden [83] and the Ivory-billed Woodpecker (*Campephilus principalis*) [84], Red-cockaded Woodpecker (*Picoides borealis*) [85], and Red-headed woodpecker (*Melanerpes erythrocephalus*) [86] in the United States. Forest-breeding raptors and other predatory birds whose declines are associated with logging include the Black Stork (*Ciconia nigra*) in the Baltic countries of Estonia, Lithuania, and Latvia [78,79] Figure 2 and the Northern Goshawk (*Accipiter gentilis*) [77] and Northern Spotted Owl (*Strix occidentalis caurina*) [80] in the northwestern United States. In the case of Black Storks, reproductive success is tied to the availability of mature trees, which have increasingly been eliminated since the intensification of forestry in the Baltic countries starting in the mid-1990s [78,79] Figure 3. In addition to cavity- and canopy-nesting birds, ground-nesting and insectivorous birds in general appear to be particularly vulnerable to the negative effects of logging in mature forests [2,4,6–9].

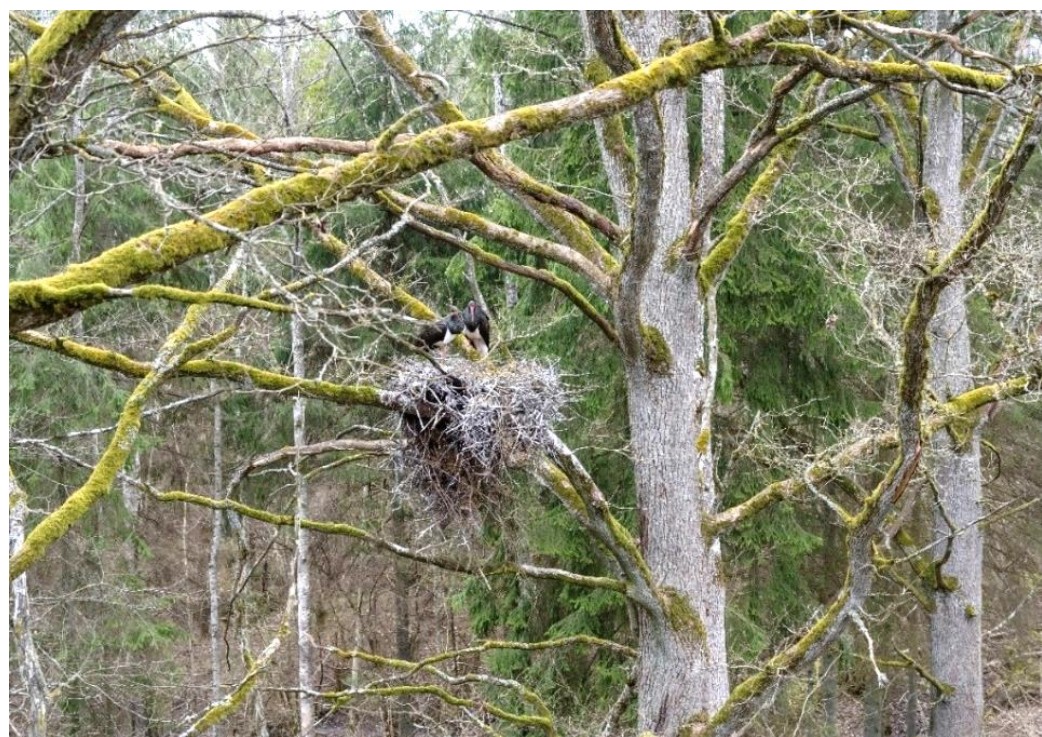

**Figure 2.** A migratory bird that depends on large, mature trees for nesting in Europe, the Black Stork (*Ciconia nigra*) has been extirpated from Scandinavia and is now critically endangered in Latvia [79]. Photo by Māris Maskalāns.

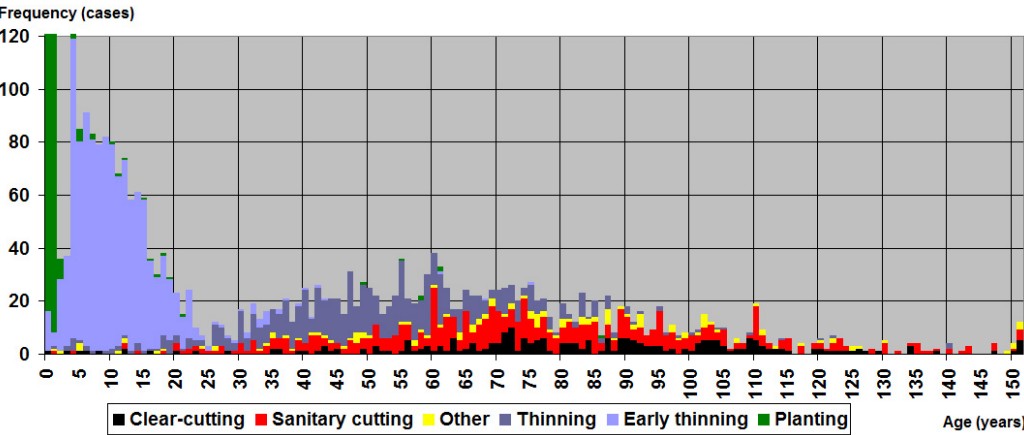

**Figure 3.** Frequency and intensity of logging operations around 117 Black Stork (*Ciconia nigra*) nests in Latvia from 1995 to 2018 show the lack of any "peace period" from logging over time, despite the existence of reserves intended to provide protection from logging operations. Data comprise 2438 forest compartments, with an average of 20.8 ha for each nest. The value of planting operations for age class 0 is 658.

An important question in interpreting forest bird population trends as indicators of biodiversity and environmental change is whether and how their responses to forest management may be separated from their responses to climate change [87,88]. Changing precipitation and temperature patterns affect forest birds directly as well as indirectly through their impacts on habitat, food resources, and other factors [87–92]. For example, Hazel Grouse populations have declined by 46% in European forests since 1980 [64], and analysis of climate conditions from 1966–2015 shows that the number of days with a sufficiently deep snow cover to enable night roosting in snowholes has been reduced by ~17 days, with implications for reduced survival [93]. Forest bird communities are

changing both in production forests and protected areas, with research suggesting that forest degradation due to timber harvest may reinforce the effects of climate change [87,88]. Bird monitoring data are thus increasingly important not only to understand the effects of ongoing forest habitat change but also climate change [2].

Many bird population trends vary spatially, as land use, forest management, and effects of climate change differ in different areas [2,78]. For example, the forest bird declines in Fennoscandia over the last three decades [2,12,64], contrast with increases in some species elsewhere in Europe, including Ireland, France, and Spain [2,64]. Likewise, Black Stork declines in the Baltic countries [78,79] contrast with increasing Black Stork numbers following earlier declines in western Europe [78]. Migratory birds that nest in the northern hemisphere must adapt to environmental changes to their breeding grounds as well as on migration routes and in wintering areas, many of which are in the tropics. Common Whitethroats (*Sylvia communis*) and American Redstarts (*Setophaga ruticilla*) are examples of species whose breeding populations are known to be influenced by conditions at their wintering grounds [2,5]. The declines of many long-distance migratory birds that nest in forests in Europe, Asia, and North America highlight the need to improve forest management not only in the northern hemisphere but also in the tropics [2].

## 3. Conservation Approaches in Natural Forests Used for Timber Production

As timber production is correlated with forest biodiversity loss, addressing the trade-offs between logging and conservation has increasingly become the focus of forest management studies [19,20]. Despite many research advances, there nevertheless remains a strong discrepancy between the forest conservation strategies applied to date and those actually needed to maintain viable populations of many species [9]. Understanding the habitat relationships and population dynamics in production forests demands consideration of the roles of forest fragmentation, edge effects, and corridors among other factors [94–102]. Uneven-aged silviculture through selective logging is among the oldest examples of the integrative use of forest resources, but its practice decreased in the second half of the 20th century in favor of more intensive, industrial logging [31,35]. In recent decades, however, rising interest in sustainable forestry and nature conservation has helped build renewed support for uneven-aged silviculture and other approaches to foster biodiversity in production forests [22,30,33].

Multi-aged forest stands have greater resilience to disturbances and similar productivity compared to even-aged stands, and thus may mitigate post-logging declines in forest biodiversity [23,24]. Continuous cover forestry, which is practiced in many areas in Europe, includes silviculture that features continuous and uninterrupted maintenance of forest cover and avoids clearcutting [32]. Retention forestry, which is practiced in many areas in North America, focuses on maintaining continuity in forest structure, function, and composition throughout logging operations and likewise avoids clearcutting [33,43]. A more recently developed approach, natural disturbance emulation disturbance aims to conduct logging operations that maintain forest structure and function in a similar manner to that which would result from natural disturbances [34,36,46,47]. In boreal forests of Canada, for example, natural disturbances have included forest fires that burn large areas, and clearcuts covering similar areas may thus partially emulate natural disturbances in this region [96]. Conservation strategies specifically targeting vulnerable species and guilds and identifying ecological thresholds can complement these approaches [8,9,11,42], as does the designation of protected reserves [103,104], as unlogged forests remain vital to sustaining many forest species and ecosystem functions [21,49,105].

## 4. Ten Principles of Bird-Friendly Forestry

Bird-friendly forestry plans should be driven by the highest conservation priority species in the management area [9,36,44,47], with a clearly articulated vision and quantifiable objectives through which success can be measured [38]. If the vision includes maintaining a breeding population of a particular woodpecker species, for example, the

current abundance of that species should be estimated based on survey data [63], together with demographic parameters such as avian productivity and survival wherever possible. These data can then serve as a baseline for continuing to monitor this woodpecker population over time, adapting management as needed to achieve the conservation objective [63,106]. To this end, we present below 10 bird-friendly forestry principles that may be customized for particular conservation objectives.

### 4.1. Protect and Enhance Vertical Structure through Uneven-Aged Silviculture

Old-growth forests harbor high structural complexity and habitat heterogeneity that is generally lacking in young successional forests; working to approximate this heterogeneity into younger stands increases their value to birds [6–8]. By avoiding cutting all the largest or most valuable trees in a single rotation, foresters can promote uneven-aged silvicultural systems that allow forests to maintain a full range of development stages after stand-replacing disturbances, including old-growth trees. Uneven-aged, selective cutting mitigates species declines compared to even-aged silviculture, as the resulting forest landscapes contain a greater range of development stages resulting from disturbance and thus support a greater range of forest birds [23,24,33]. The protection and enhancement of vertical structures can be achieved via the use of selective logging targeting single trees and small stands. In even-aged stands that lack structural heterogeneity, harvesting larger stands of trees intensely may be necessary to avoid short-term financial loss. In such cases, uneven-aged stands can be achieved over time by promoting early successional forest habitat through patch cutting to benefit a range of bird species.

### 4.2. Leave Dead and Dying Trees and Coarse Woody Debris in Different Decay Stages

Large snags (standing dead or dying trees) and coarse woody debris (fallen dead or dying trees, logs, branches, and other remnants) provide important bird habitat features and support invertebrate communities that are an important food source for many birds [21,76,81,82,107] Figure 4. Historically, standing dead wood in unlogged European forests could comprise 20–50% of standing trees [82]. Avoiding disturbance to coarse woody debris may allow a higher proportion of resident species to survive the regeneration phase at the stand level [48]. When safe to do so, foresters can support bird conservation by keeping standing and fallen dead and dying trees, ideally including at least >10 snags/ha, at least one snag > 45 cm diameter at breast height (dbh), and at least 8 > 30 cm dbh [48]. Even small increases in available dead wood may benefit birds in production forests. For example, an increase in standing dead wood to 3.2% of the total standing trees in Swiss forests was correlated with an increase in Middle Spotted Woodpeckers [82].

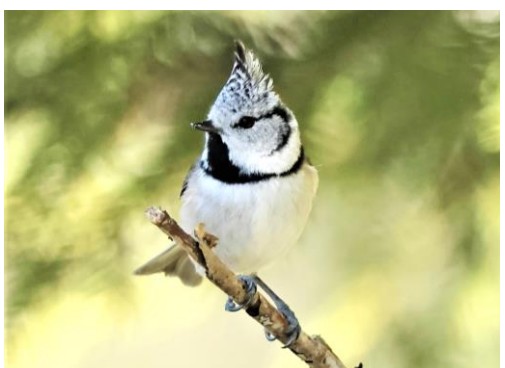 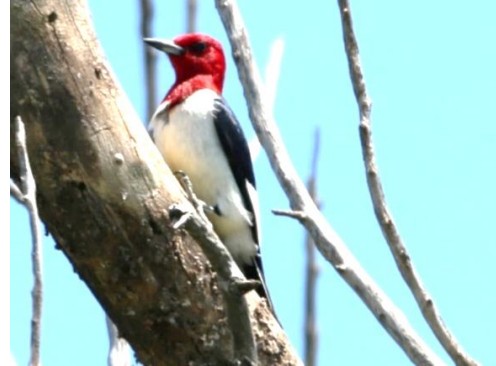

**Figure 4.** Leaving dead wood benefits cavity-nesting birds such as the Crested Tit (*Lophophanes cristatus*; left), which has declined by 53% since 1980 in Europe [64], and Red-headed Woodpecker (*Melanerpes erythrocephalus*; right), which has declined by 67% since 1970 in North America [66]. Crested Tit photo by Nils-Fredrik Nilsson; Red-headed Woodpecker photo by Nico Arcilla.

### 4.3. Maintain Residual Large, Green Trees

Retaining large (>55 cm dbh) mature trees in production forests improves habitat for disturbance-phase insects and birds in regenerating stands [43,53,82,98,99,108]. High biodiversity is characteristic of boreal forests that grow to the climax stage, which occurs about ~50–100 years later than when many tree species are typically harvested [7]. For example, most Black Storks nest in trees that are over twice as old as the age when most trees of their species are logged (41–101 years) [Table 1]. Conserving legacy trees and allowing regeneration of longer-lived species partially emulates the natural successional pattern following logging operations [48]. Maintaining at least 50 large, mature trees/ha, as well as larger retention tree groups in species-rich areas can mitigate biodiversity and habitat loss [48]. Remnant forest stands may be left because of extraordinary properties and/or lower economic interest, such their location on rocky outcrops or steep slopes [36]. Insufficient mature tree retention in Scandinavian production forests has been associated with regional declines in many forest bird species [12]. Particular tree species, such as large oaks (*Quercus* spp.), are important for birds such as the Middle Spotted Woodpecker, for example, whose occupancy, foraging, nesting, and juvenile dispersal are positively correlated with the density of large oaks [82]. Wherever possible, foresters should thus retain oaks and other nut- and fruit-bearing trees such beech, pine, and cherry.

**Table 1.** Average ages of nesting tree species used by Black Storks (*Ciconia nigra*) in Latvia, compared to the average ages at which the same tree species are logged.

| Species | Average Age | Minimum Age | Maximum Age | Logging Age | Sample Size |
|---|---|---|---|---|---|
| Pine (*Pinus sylvestris*) | 206 | 81 | 430 | 101 | 455 |
| Oak (*Quercus robur*) | 169 | 135 | 190 | 101 | 30 |
| Aspen (*Populus tremula*) | 100 | 70 | 135 | 41 | 252 |
| Black Alder (*Alnus glutinosa*) | 120 | 85 | 159 | 71 | 26 |
| Spruce (*Picea abies*) | 157 | 119 | 210 | 81 | 17 |

### 4.4. Establish and Maintain Uncut Reserves, Ideally Connected by Corridors

Protected areas play an essential role in conservation, and many forest-nesting bird species appear better able to adapt to changing climatic conditions in protected areas than outside them [87,109]. Leaving uncut reserves for species that prefer or require mature, closed-canopy forests provides a refuge from logging operations and mitigates the landscape-scale effects of logging on birds [103,104,109]. Old-growth stands serve as crucial breeding, nesting, and foraging habitat for many bird taxa, including woodpeckers, raptors, storks, and grouse [7–9,62,110], with larger reserves benefiting more species. For example, in northeastern forests in the United States, > 20 ha stands exhibit the increased bird diversity compared to smaller stands, including species such as the Scarlet Tanager (*Piranga olivacea*), Black-throated Green (*Setophaga virens*) and Black-throated Blue Warbler (*Setophaga caerulescens*), and Eastern Wood-pewee (*Contopus virens*) [48]. In addition, negative responses to logging are generally smaller in connected fragments compared to isolated fragments [111,112]. Many forest species have a negative relationship with edges and fragmentation, and therefore may be consistently absent from small forest patches or corridors [111,112]. For example, Cerulean Warblers may be absent in forest patches < 138 ha and Worm-eating Warblers (*Helmitheros vermivorus*) may be absent in forest patches < 21 ha in the eastern United States [111]. Forest fragmentation may introduce threats to many birds due to negative edge effects such as increased predation and brood parasitism [113,114], and may be associated with lower nestling body mass due to lower feeding rates [115–123].

Negative edge effects on birds can be partially mitigated in part by developing irregularly shaped edges, also known as "softening" forest edges from straight to irregular lines [115].

### 4.5. Maximize Forest Interior through Retaining Large Contiguous Forest Tracts

Forest interior may be defined as habitat that occurs in unbroken forest > 100 from forest edge [116]. The majority of vertebrates are either positively or negatively affected by forest edges, with forest interior species increasing in abundance 200–400 m from forest edge [117]. Maintaining large areas of contiguous forest helps increase the amount of forest interior available to birds and mitigate the effects of forest fragmentation and isolation [74]. Forest remnants sufficiently large to include an average songbird territory (>5 ha) may be inadequate for many bird species when embedded in a clearcut dominated landscape [48]. In Germany's Black Forest, bird species richness and diversity in 1 ha forest patches increased with increasing proximity to other forest patches [121], highlighting the importance that landscape configuration has for bird conservation in temperate montane forests in this region. Conversely, forest degradation caused by logging can also lead to changes in the occurrence of nesting songbirds, not only in logged stands but also in surrounding, untouched stands [122]. In addition, while forest corridors may be conducive to retaining birds in production forests, such corridors may be unlikely to offset the negative impacts of fragmentation on the abundance, productivity, and survival of many species [112]. Moreover, in coniferous forests in the northwestern United States, the amount of available habitat appeared to be more important for birds than its connectivity and configuration [118]. Precise area requirements will vary by species and region [108], but maintaining forest stands as a patchwork mosaic of different structural stages across broad landscapes will protect foraging and nesting sites for many forest bird species [8].

### 4.6. Maintain Buffers around Streams, Rivers, Wetlands, and Known Nesting Areas

Many forest breeding species require riparian buffers in order to nest successfully [120], and foresters can allow them to do so by maintaining such buffers where logging operations take place. Buffer sizes will depend on the conservation needs of focal species. For example, Louisiana Waterthrush (*Parkesia motacilla*), migratory birds that nest in North American temperate forests, require > 40 m buffers of closed-canopy forest along contiguous > 1.5 km stream networks to breed successfully [123]. Many other forest-breeding birds benefit from buffers around their nesting areas and may either avoid nesting in areas with forestry activities or exhibit decreased reproductive success. For example, Cinerous Vultures (*Aegypius monachus*) avoid forestry activities within 300 m of their nests [110], and protected buffers at least 500 m around Black Stork nests is associated with a greater probability of nest success compared to nests exposed to disturbances within 100 m [79]. Many forest interior species also have lower relative abundance or territory density near roads [124], and whenever possible, forestry roads should be closed to vehicular traffic following logging operations to allow forest to regenerate [49].

### 4.7. Maintain Horizontal Stand Structure and Enhance Vegetation Diversity in Canopy Gaps

Bird species richness, diversity, and abundance benefit from natural disturbances that create light gaps, which are associated with higher species richness, diversity, and breeding bird abundance compared with closed forests [107,125]. Silvicultural practices that create small gaps appear to benefit some forest species without negatively influencing others [14,57]. Encouraging such patterns of forest regeneration after disturbances can be used to enhance forest composition and structure [25]. In forest stands exhibiting mid- to late successional structures, foresters may improve bird conservation through gap sizes ranging from the crown area of a single large tree up to 0.2 ha, with gaps > 0.1 ha admitting enough light to benefit some commercially important, shade-intolerant tree species [48]. Foresters can support both old and young growth in forests by allowing such light gaps to form via small-scale disturbances and gap dynamics resulting from dead single or small stands of trees.

## 4.8. Extend the Temporal Scale of Logging Cycles through Prolonged Rotations

Increasing the cutting age of retention trees can benefit many bird species, particularly those that require mature trees for nesting and foraging, including both residents and migrants [11,49,79,108]. For many tropical forest bird species that decline following logging operations, for example, 40 years is insufficient time to allow recovery [11]. In boreal forests of Québec, bird species richness and community composition in regenerated forest 60–70 years after clearcuts stands still differed from those comparable stands that had not been logged, in spite of the fact that stand structure characteristics became similar after 40 years, suggesting that longer logging rotations will improve their bird conservation value [108]. In temperate forests of Oregon, extending 40–50 year rotations to longer cycles may reduce conflicts between timber production and environmental, aesthetic, and wildlife values without reducing long-term timber production [126].

## 4.9. Minimize Disturbance to Forests after Logging and during the Bird Breeding Season

Whereas natural disturbances such as forest fires are typically followed by ecological succession without human disturbance, logging operations may be followed by a high intensity of human activities, with corresponding detrimental impacts on birds [49,80,96]. Hunting, recreation, and other human activities following logging operations, may drive declines in bird populations that could otherwise recover over time [49]. Protecting forests from detrimental human activities following logging operations and scheduling logging operations to take place outside the main bird breeding season can help reduce their negative impacts, including allowing nesting birds to successfully fledge young. For example, Black Stork nests disturbed in April all failed to produce any fledglings, whereas despite disturbances, at other times, 50% of pairs successfully produced young [79]. Bird conservation initiatives for grassland birds have found landowners willing to avoid harvesting hay during the bird breeding season to protect nesting birds and chicks [127]. Likewise, forest landowners in Finland have voluntarily avoided logging during the bird breeding season to protect nesting raptors including European Honey Buzzards (*Pernis apivorus*), Common Buzzards (*Buteo buteo*), and Northern Goshawks [128].

## 4.10. Manage for Focal Species and Guilds

Conservation strategies should focus on birds known to exhibit negative responses to logging operations, including many ground-nesting, cavity-nesting, canopy-nesting, predatory, and insectivorous species, including grouse and other gamebirds, raptors, woodpeckers, hornbills, and songbirds, among others [2,5–11,21,46] Figure 5. For example, managing for Black Storks in Latvia would include reducing the intensity of logging operations, extending rotation periods, and retaining mature trees used for nesting as well as surrounding forest within at least 500 m of nests [79]. Managing for the Louisiana Waterthrush in the southeastern United States would include maintaining > 40 m buffers of closed-canopy forest along contiguous > 1.5 km stream networks [123]. Forest management plans targeting conservation priority focal species or guilds would be expected to benefit many other bird taxa with similar life histories [11,49].

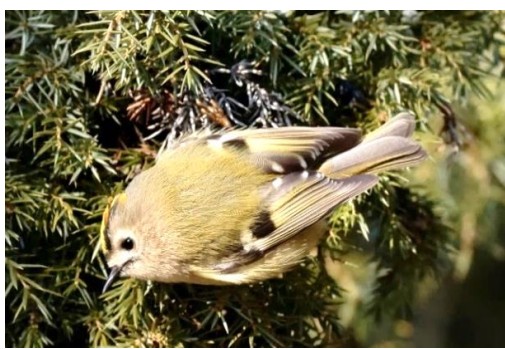 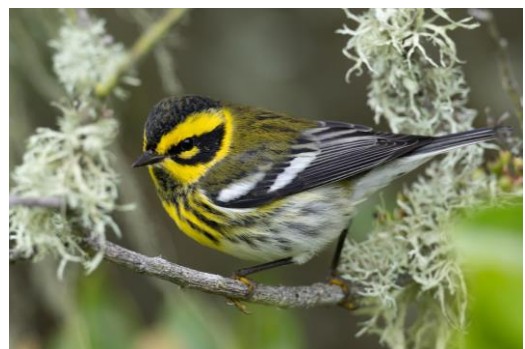

**Figure 5.** Examples of declining songbirds that require mature temperate and boreal forests to nest include the Goldcrest (*Regulus regulus*; left), which has declined by 49% since 1980 in Europe [64], and Townsend's Warbler (*Setophaga townsendi*; right), which has declined by 15% since 1970 in North America [66]. Goldcrest photo by Nils-Fredrik Nilsson; Townsend's Warbler photo by Dan Marks.

## 5. Conclusions and Future Directions

Bird-friendly forestry can make important contributions to conservation, as most of the world's natural forests are subject to logging operations [1], and their management has critical consequences for wildlife [22]. To improve the bird conservation value of production forests, any or all of the principles we present here may serve as a starting point toward developing bird-friendly forest management plans [Table 2]. As birds interact with changing habitats and are influenced by the landscape context, conservation strategies and focal species must be area-specific, clearly articulated, and have measurable objectives [38]. Data from forest bird monitoring programs should inform forest management plans customized for particular areas and species [2,63,64].

**Table 2.** Ten principles and practices of bird-friendly forestry.

| Principle | Practice | References |
|---|---|---|
| 1. Protect and enhance vertical structure | Employ uneven-aged silviculture | [8,9,23,24,28,29,33] |
| 2. Leave dying and dead trees and coarse woody debris | Retain snags, avoid crushing logs, and scatter tops and limbs | [7,8,21,81,82,107] |
| 3. Maintain residual large green trees | Protect mature trees and stands | [43,53,98,99,108,128] |
| 4. Integrate conservation areas into production forests | Establish and maintain uncut reserves and corridors | [62,87,103,104,109] |
| 5. Maximize forest interior and minimize fragmentation | Retain contiguous stands and reduce and "soften" edge areas | [6–8,74,102,112,119] |
| 6. Maintain buffers around riparian areas and nests | Protect riparian and nest sites | [6–8,79,120,123] |
| 7. Maintain horizontal stand structure | Establish small canopy caps | [14,25,57,94,107,125] |
| 8. Lengthen logging rotations | Increase cutting age of trees that have high value to birds | [11,40,49,79,108,125] |
| 9. Minimize post-logging and breeding season disturbance | Allow logging roads and trails to regenerate forest | [79,80,96,105,127,128] |
| 10. Manage for focal species and guilds | Identify priority species and quantitative objectives | [5–11,21,42,64,66,71] |

Future research should target knowledge gaps in forest bird habitat relationships and management strategies [4,6–8], including defining the amount of forest habitat required by focal species to maintain or increase their populations, and evaluating the effects of non-native timber plantations, non-native predators, insect infestations, forest diseases, and other threats on forest birds. Much past research has taken place over small temporal scales and/or small spatial scales, and there is a need for more landscape studies, which are rare and a relatively recent phenomenon [6,7,104]. Research priorities include evaluating trade-offs between conservation areas paired with more intensive management (land sparing) and bird-friendly forestry approaches across larger areas to meet wood production

demands (land sharing) [129]. Ongoing research is needed to better understand of the effects of changing forest management and climatic conditions on forest bird populations, particularly those exhibiting declines [64,66,72,87].

Recent reviews suggest that many boreal and temperate species are steeply declining, and therefore should be monitored with attention to the effects of forestry and conservation strategies [2,66,106]. Tropical forest bird resident species merit particular attention because many of them are very poorly known; the effects of logging operations on their populations may be devastating, and major improvements in tropical forestry practices are urgently needed [11,15,49]. To this end, investigating the roles of incentives programs and legislation are needed to encourage bird-friendly forestry, both on state and private lands. Increasing systematic bird sampling and expanding monitoring programs into new areas will help make valuable contributions [2]. Finally, future studies evaluating the outcomes of forest management on bird population dynamics and trends [130], and specifically the effects of various conservation approaches highlighted here, will provide critical empirical data to inform adaptive management.

**Author Contributions:** Conceptualization and investigation, N.A. and M.S.; writing—original draft preparation, N.A.; writing—review and editing, N.A. and M.S.; funding acquisition, M.S. and N.A. All authors have read and agreed to the published version of the manuscript.

**Funding:** Donors to the International Bird Conservation Partnership (IBCP) provided funding to conduct this review. The Baltic-American Freedom Foundation (BAFF) funded a 2018 symposium at the University of Latvia entitled "Forestry and Biodiversity: International Perspectives on Trade-offs, Problems, and Solutions," which stimulated discussions that contributed to this paper.

**Institutional Review Board Statement:** Not applicable.

**Data Availability Statement:** Data used for this review may be accessed through the respective citations provided and/or by contacting the authors.

**Acknowledgments:** We thank the administration, faculty, and staff at the University of Latvia, Riga, and to the Baltic-American Freedom Foundation for supporting the above-mentioned symposium that originally inspired this paper, and to all who attended and participated. This paper has also been informed by forest bird conservation symposia and discussions at the 2017 and 2019 meetings of the European Ornithologists' Union. Editor Jukka Jokimäki, three anonymous reviewers, Per Angelstam, Robert J. Cooper, and Ola Svensson provided helpful feedback that allowed us to improve earlier drafts of this manuscript. We are very grateful to Dan Marks for providing photographs in Figures 1 and 5 (Wilson's Warbler and Townsend's Warbler), Māris Maskalāns for providing the photograph in Figure 2 (Black Stork), and Nils-Fredrik Nilsson for providing the photographs in Figures 3 and 5 (Crested Tit and Goldcrest). Finally, we thank donors to IBCP for their support, which made this review possible.

**Conflicts of Interest:** The authors declare no conflict of interest.

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
