# Peer review of "Ten Principles for Bird-Friendly Forestry: Conservation Approaches in Natural Forests Used for Timber Production"

_2673-6004, doi:10.3390/birds4020021_

Round 1

Reviewer 1 Report

General Comments

In this paper the authors present 12 principles for bird-friendly forestry to promote conservation in a forest management context. Although I agree with the general premise that forest management could be improved to support forest birds I felt that the paper lacked a recognition of the variation in ecosystems. The paper failed to articulate a consistent scope, is it global? Is it boreal? Is it Latvia? The authors also did not convince me that they had a good understanding of the concept of natural disturbance emulation. Or the reality that natural disturbance emulation is not entirely consistent with their 12 principles. Lastly it seemed that the authors had very strong opinions about the failing of current forest management and did not recognize that current sustainable forest management both recognizes the need for biodiversity conservation and strives to practice it through the application of guidelines for natural disturbance emulation. This is not just a concept discussed by scientists, it is a policy supported by legislation in Canada at least but probably elsewhere. Also there is not a lot of empirical evidence linking forest management to decline of SAR in boreal Canada. I cannot speak to other regions, but the citation evidence provided here was not strong. I am not suggesting that forest management is not having an impact, only that we need to be rigorous in our analysis of the empirical evidence. My feeling is that the authors of this paper were not rigorous enough and that that undermined my confidence in their conclusions.

Specific Comments

Explain why you use citations with author’s names but then order the references by some random? Order. I can’t check the references. Is this a journal standard or did the authors do something wrong?

Lines 65-70. The authors highlight boreal forests and include North American work so it is necessary to acknowledge that often in fire prone boreal forests in Canada at least, the fires are large and stand replacing rather than killing individual or small groups of trees.  In fire prone boreal forests clearcutting is considered to be natural disturbance emulation. And yes in general species richness is highest with moderate disturbance but that does not at all reflect the philosophy of natural disturbance emulation. The goal is not to maximize richness but to maintain the natural community composition.

Lines 58-60. In my experience this is not at all true. Sustainable forest management (SFM) as practiced in Canada and I am sure elsewhere, uses ecosystem management principles designed to maintain forest wildlife. This is one of the major drivers of current forest policy. 

Lines 107-111. The authors need to be much more critical in their use of citations. There is not strong evidence that decline in Canada Warbler, Evening Grosbeak, Rusty Blackbird and Olive-sided Flycatcher are the result of unsustainable forest management. These kinds of blanket statements undermine the trust the reader will have in your paper. Your argument that bird friendly forest management practices is sound even without these unfounded claims. I refer the readers to the Environment Canada Endangered species status reports (COSEWIC) which can be found on line but also Cornell’s Birds of the World for credible discussions on the conservation issues for these species.  Some information can also be found in a 2014 review of natural resource development including forestry on biodiversity in Canada’s boreal forest (Venier et al 2014, Environmental Reviews).

Lines 157-158. Why single out the US on this? This sentence seems out of place.

Also is this paper focused on the boreal or more broadly relevant. Its not clear from the text yet.

Section 3 on Black Stork seems out of place. The paper starts out at a global scale but then switches to this very national level issue for Latvia. But then switches back to natural disturbance eemulation and 12 principles. This is not a good flow. Perhaps you could work in the Stork example in section 5.

Line 248 needs a citation

Line 255-263 This needs a citation…in eastern Canada where” ? Is this boreal. You have quoted someone but not cited them. The quote that you use does not support natural disturbance emulation. Just the opposite “complement not duplicate natural disturbance”  who said this?

Bird friendly forestry does not need to be entirely about natural disturbance emulation. There are fine-filter approaches that would complement as the quote says.

Line 299 this is far too specific a recommendation for a ‘global’ paper and does not reflect the large heterogeneity of global ecosystems

Lines 324…certainly not boreal species.

Line 352-355 does Louisiana Waterthrush require large tracts and large trees? Is it a boreal species?

Line 375 needs citation

Lines 457-459 in what direction?

Line 385-386…not universally true where is the citation and where does this apply

Lines 485-493 the authors are introducing a completely new idea in the conclusion

Author Response

Thank you for your review of our manuscript, which we have used to substantially revise the paper. Below please find our specific responses to your feedback, in italics to contrast it with your comments.

In this paper the authors present 12 principles for bird-friendly forestry to promote conservation in a forest management context. Although I agree with the general premise that forest management could be improved to support forest birds I felt that the paper lacked a recognition of the variation in ecosystems. The paper failed to articulate a consistent scope, is it global? Is it boreal? Is it Latvia? The authors also did not convince me that they had a good understanding of the concept of natural disturbance emulation. Or the reality that natural disturbance emulation is not entirely consistent with their 12 principles. Lastly it seemed that the authors had very strong opinions about the failing of current forest management and did not recognize that current sustainable forest management both recognizes the need for biodiversity conservation and strives to practice it through the application of guidelines for natural disturbance emulation. This is not just a concept discussed by scientists, it is a policy supported by legislation in Canada at least but probably elsewhere. Also there is not a lot of empirical evidence linking forest management to decline of SAR in boreal Canada. I cannot speak to other regions, but the citation evidence provided here was not strong. I am not suggesting that forest management is not having an impact, only that we need to be rigorous in our analysis of the empirical evidence. My feeling is that the authors of this paper were not rigorous enough and that that undermined my confidence in their conclusions.

Thank you for highlighting these issues. We have now revised the paper to give more attention to the scope in lines 107-110, “We drew on an abundance of studies of bird-forestry relationships, particularly in boreal and temperate forests in North America and Europe, where the majority of such studies have taken place [2, 8, 9, 39, 40, 41, 42, 43, 44, 45, 46, 47, 48], as well as studies in tropical forests.”

We have now provided more information and references about natural disturbance emulation, for example in lines 264-267, “A more recently developed conservation approach in production forests, natural disturbance-based management is intended emulate disturbance effects with management and build variable treatment intervals (cutting cycles/harvest rotations) into management regimes [34, 36, 46, 47].”

We have also included a statement to acknowledge your point about the recognition of the need for sustainable forest management, including in lines 64-66: “Addressing the declines of birds and other biodiversity associated with logging in natural forests is a goal towards which much research has been directed, together with legislation and incentives in many countries and regions [2, 6, 7, 8, 9, 15].”

In addition, we have included more empirical evidence linking forest management to forest species declines, especially from northern Europe, where forest management has been more intense and most data on declines has been documented, including in lines 54-64, “Correspondingly, monitoring data show that forest bird populations in Norway, Sweden and Finland have been declining for the last three decades [2], with the expanding timber frontier has led to the elimination of unlogged forest in Finland [1] and the de-clines of the last intact old-growth forests in Sweden [12]. Forestry operations are also having impacts on Canadian forests and their birds [4], such as the large-scale conversion of southern Québec’s boreal forest from mixed to deciduous forest [13, 14]. 14]. Moreover, the majority (50-60%) of remaining tropical forests, which harbor the world’s greatest levels of terrestrial biodiversity, are dedicated to timber production, and corresponding declines in forest bird species have been documented [10, 11, 15].”

We hope the increased inclusion of empirical data, including additional information and references in lines 161-197, and references helps bolster confidence in our conclusions. Further specific responses are below.

Specific Comments

Explain why you use citations with author’s names but then order the references by some random? Order. I can’t check the references. Is this a journal standard or did the authors do something wrong?

We have now corrected the citation style and reference list.

Lines 65-70. The authors highlight boreal forests and include North American work so it is necessary to acknowledge that often in fire prone boreal forests in Canada at least, the fires are large and stand replacing rather than killing individual or small groups of trees.  In fire prone boreal forests clearcutting is considered to be natural disturbance emulation. And yes in general species richness is highest with moderate disturbance but that does not at all reflect the philosophy of natural disturbance emulation. The goal is not to maximize richness but to maintain the natural community composition.

We have now integrated this point into the paper, including the following statement in lines 264-270, “A more recently developed conservation approach in production forests, natural disturbance-based management is intended emulate disturbance effects with management and build variable treatment intervals (timber harvest rotations) into management regimes [34, 36, 46, 47]. In boreal Canada, for example, natural disturbances have included forest fires that burn large areas, and clearcutting covering similar areas may thus partially emulate natural disturbances in this region [96], maintaining bird community composition.

Lines 58-60. In my experience this is not at all true. Sustainable forest management (SFM) as practiced in Canada and I am sure elsewhere, uses ecosystem management principles designed to maintain forest wildlife. This is one of the major drivers of current forest policy. 

We have now modified the previous statement and added information about SFM approaches in lines 236-297, “Bird conservation approaches in natural forests used for timber production.”

Lines 107-111. The authors need to be much more critical in their use of citations. There is not strong evidence that decline in Canada Warbler, Evening Grosbeak, Rusty Blackbird and Olive-sided Flycatcher are the result of unsustainable forest management. These kinds of blanket statements undermine the trust the reader will have in your paper. Your argument that bird friendly forest management practices is sound even without these unfounded claims. I refer the readers to the Environment Canada Endangered species status reports (COSEWIC) which can be found on line but also Cornell’s Birds of the World for credible discussions on the conservation issues for these species.  Some information can also be found in a 2014 review of natural resource development including forestry on biodiversity in Canada’s boreal forest (Venier et al 2014, Environmental Reviews).

We have revised the examples of bird species declines linked with forest management and added a paragraph including additional information on indicators, including COSEWIC, in lines 136-157, “Forest birds as indicators of management practices and environmental change.” We have now included and referenced the informative Environmental Reviews paper by Venier et al. 2014 (reference 14).

Lines 157-158. Why single out the US on this? This sentence seems out of place.

We have revised this sentence and do not single out the US in the revision.

Also is this paper focused on the boreal or more broadly relevant. Its not clear from the text yet.

We draw largely on boreal and temperate forestry research, as mentioned above, and also include some references to tropical forest research, as discussed in the revision.

Section 3 on Black Stork seems out of place. The paper starts out at a global scale but then switches to this very national level issue for Latvia. But then switches back to natural disturbance eemulation and 12 principles. This is not a good flow. Perhaps you could work in the Stork example in section 5.

Line 248 needs a citation

We have revised this sentence and added appropriate citations.

Line 255-263 This needs a citation…in eastern Canada where” ? Is this boreal. You have quoted someone but not cited them. The quote that you use does not support natural disturbance emulation. Just the opposite “complement not duplicate natural disturbance”  who said this?

This statement was included in error and has been removed.

Bird friendly forestry does not need to be entirely about natural disturbance emulation. There are fine-filter approaches that would complement as the quote says.

Agreed; we have revised the text in an effort to reflect this.

Line 299 this is far too specific a recommendation for a ‘global’ paper and does not reflect the large heterogeneity of global ecosystems

We have revised this sentence.

Lines 324…certainly not boreal species.

We have revised this sentence.

Line 352-355 does Louisiana Waterthrush require large tracts and large trees? Is it a boreal species?

Louisiana Waterthrush require riparian forest buffers along streams to nest successfully and we have now revised the text to indicate this; it is a temperate forest breeding species and Neotropical migrant.

Line 375 needs citation

We have revised this sentence and added appropriate citations.

Lines 457-459 in what direction?

We have revised this sentence and added appropriate citations.

Line 385-386…not universally true where is the citation and where does this apply

We have revised this sentence and added appropriate citations.

Lines 485-493 the authors are introducing a completely new idea in the conclusion

We have revised the conclusion accordingly. Thank you again for many helpful comments, questions, corrections that helped us improve this manuscript.

Reviewer 2 Report

Please, see attached file with comments. I hope they will be usefull.

Overall, this is an interesting piece but requires additional work to improve the narration and provide proper references to some of the statements.

Author Response

Thank you for your helpful comments. Please find our responses attached.

Reviewer 3 Report

Dear Authors

Please find my comments attached.

Author Response

Title and authors:

Twelve principles for bird-friendly forestry: conservation approaches in natural forests used for timber production

Nico Arcilla 1,2* and Maris Strazds 3

But:

Citation: Arcilla, N.; Strazds, M.; Rosamond, K. M., Angelstam, P. Twelve principles for bird-friendly

forestry: conservation approaches in natural forests used for timber pro-

duction. Birds 2022, 3,

Which one is the right one?

>> Response: We have corrected the citation to match the author list – thank you!

General comments

The manuscript is a nice, well-written review on an important topic. However, it might be a little bit more compact and more catchy. The figures are only loosely connected to the text. Figure 1 should be commented and explained in more detail. The photos are not necessary but if wanted they may serve as decoration with some relevant additional information in the legends. This information could also explain why the status of the example species has changed. In addition, there could be a summary-like table that shortly presents with one or a few representative references the mentioned principles for bird-friendly forestry in some logical order. Of course, the order should be the same as in the main text.

>> Response: Thank you. We have now used reviewer feedback to reorganize and rewrite the manuscript, including condensing the 12 principles to 10 as part of as effort to make it more focused. We have added more detail to the caption of Fig 1 from the original text, which now appears as Fig 4. The figures with photographs remain simply as examples of declining forest birds, and now feature information on their trends in the captions, but can be removed on request. We have now listed all 10 principles in the abstract and simple summary for easy reference; we have not added a table in case this appears to be repetitive but can do so at the editor’s request, or can make a graphical abstract using such a table as the main theme.

Minor comments

Line

38        In addition to              ??

61-62 (Mon-

kkonen et al. 2014;     >

(Monkkonen et al. 2014; >     (Monk-

konen et al. 2014;

82   may intensifies climate change's        >          may intensify climate change's

176 >> Figures

Figure 1          The reference in the text is missing. Maybe there is no use to number the photos if they are not ref erred in the text.

245      example, Logging       >            example, logging 246 > Scientific names of species in italics

370      may createa range       >          may create a range

  1. 9-10 (etc) Add the scientific names of the species.

403      5.9 subtitle

416      5.9 subtitle

5.11 Missing

472      4.         >         5.

>> Responses: Thank you very much for pointing out these errors; we have noted them and sought to correct them in the revised manuscript submitted here.

Round 2

Reviewer 2 Report

Please, see comments on attached file.

Author Response

Thank you very much for your careful reading of our manuscript and your corrections and feedback. We have implemented almost all of your suggestions; please find our point-by-point responses in the attached file.
